# Triple-band highly sensitive terahertz metamaterial absorber for biomedical sensing applications

**Asad Miah**[ID]*, **Sams Al Zafir, Md. Hasnain, Joyonta Das, Sayed Muhammad Anowarul Haque, Abdul Wahed**

Department of Electrical and Electronic Engineering, Mymensingh Engineering College, Mymensingh, Bangladesh

* asad.mec.eee@gmail.com

**citation:** Miah A, Al Zafir S, Hasnain M, Das J, Anowarul Haque SM, Wahed A (2025) Triple-band highly sensitive terahertz metamaterial absorber for biomedical sensing applications. PLoS One 20(8): e0328077. https://doi.org/10.1371/journal.pone.0328077

**Data availability statement:** All relevant data are within the paper and its Supporting information files.

## Abstract

Due to the growing interest in metamaterials for biomedical applications, this study presents the design and analysis of a novel, compact, triple-band metamaterial absorber for biological sensing in the terahertz range. The structure, with dimensions of $41 \times 41$ $\mu m^2$, exhibits exceptionally high absorption rates above 99% at three distinct resonance frequencies of 1.85, 3.62, and 5.63 THz. To validate the design, an equivalent circuit model was created and evaluated alongside electric field, magnetic field, and surface current distributions. A key focus of this work is the sensor's performance, evaluated by introducing a sensing layer with varying refractive indices. The absorber demonstrates outstanding sensitivity values of 1.5 THz/RIU, 1 THz/RIU, and 0.66 THz/RIU at the third, second, and first resonances, respectively, indicating highly reliable sensing performance across multiple frequency bands. Furthermore, its suitability for biochemical applications was evaluated by testing its ability to detect different samples, such as glucose, malaria, and cervical cancer cells. The absorber also demonstrates strong performance metrics, with a maximum quality factor of 39.13 and a figure of merit (FoM) of 6.95, supporting its reliability. Additionally, its suitability for microwave imaging (MWI) technology is also examined. The combination of near-perfect absorption and high sensitivity makes this compact metamaterial absorber a promising candidate for advanced biomedical sensing and diagnostic technologies.

## Introduction

A metamaterial is a synthetic substance with characteristics not typically present in natural materials. These properties come from their structure rather than their chemical properties. As a result, the number of studies in this field is growing regularly. Researchers are already using metamaterials to study numerous fields, including perfect absorbers [1], oil sensors [2,3], refractive index sensors [4], biosensor [5], imaging [6], and solar energy harvesting [7,8]. In recent years, there has been a surge in research on MTM-based sensing due to its small size and excellent performance. In [9], an absorber based on the DNG MTM was suggested for use in microwave sensing applications. They reported a high q-factor value of

**Funding:** The author(s) received no specific funding for this work.

**Competing interests:** The authors have declared that no competing interests exist.

1413.29 and a high absorption value of 99.9%. The detection applications for several oils are also demonstrated in this study, and the sensor can effectively distinguish between these oils. For the detection of gasoline and oil contamination, an MTM sensor based on a star-enclosed circular SRR is suggested [10]. They conducted sensing performance tests using a variety of lubricants and fuels and discovered an average frequency shift for these samples. Nonetheless, they reported that the q factor, fom, and sensitivity values were good, at 430, 1.99, and 855.70, respectively. A microwave sensor based on SRR was presented in [11] for a variety of applications. Using a variety of chemical solvents, including ethanol, methanol, and glucose, among others, they tested the sensor's performance and observed a significant frequency shift for various sample types.

Due to their potential, metamaterial research is also expanding in the biomedical field. Many researchers have evaluated their sensing abilities in the range of 1.3 to 1.39 because biological samples have refractive index values in this region [12]. For use in biomedical applications, a dual-band square-shaped MMA in the THz region was suggested in [13]. They discovered that both of their peaks had strong absorption and good sensitivity values. Additionally, this study suggested several production techniques for micro-level structures. In [14], a proposal for an ultrathin MMA for the RI detection of biomedical materials was proposed. In this study, the absorption value is 99.7%, and the FWHM and Q factor are 5.4 GHz and 19.57 GHz, respectively. They have a fom value of 3.48 and a sensitivity of 3.48 GHz/RIU, which is low.

A sensor based on metamaterials is also used to identify cancerous cells. Since almost 80% of women in the least developed nations have cervical cancer, it is the second most common form that affects women globally [15]. The mortality rate of this cancer may decrease with early identification. However, the diagnostic method of this detection system is expensive and time-consuming. An MTM-based sensor in the THz region may reduce this barrier. HeLa cells have already been used in numerous studies in this area to identify them in their early stages. A THz-based MTM biosensor was proposed in [16] for the early detection of cancer. To distinguish between cancerous and healthy cells, they employ imaging techniques. In [17], a triple band MMA was suggested for the identification of cervical cancer HeLa cells in their early stages. In the THz region, this investigation revealed three absorption peaks, all of which exceeded 99% and had acceptable quality factors. Their sensor utilises the refractive index to distinguish between cancerous and normal cells, and they also employ imaging technology to achieve this. Given that 250 million people globally contract malaria each year, prompt screening is crucial [18]. Using a metamaterial-based sensor to distinguish between healthy and malarial cells can save money and time. Numerous studies have been conducted in this field, and several types of sensors have been suggested to use the refractive index value to distinguish between malarial and healthy cells. [19] suggested a modified dual T-shaped MMA-based sensor for a variety of biological tests. These authors reported excellent and moderate absorption values at a peak frequency of 6. Next, various refractive indices are used to identify cells afflicted by malaria and to identify the presence of glucose in water. For applications involving refractive index sensing, a dual-band MMA in the THz region was proposed [20]. They tested a variety of materials, including glucose and malaria, for sensing tests, and the results were positive. For energy harvesting and stealth technology applications, a square-shaped water MTM absorber was suggested [21]. The absorber showed absorption value of more than 90% from frequencies ranges 10.4 to 30 GHz. A triple-band mtm absorber has been proposed for use in biological sensing, filtration, etc. [22]. All of the peaks exhibit good absorption values of 98%, 91%, and 98%, and these peaks also have good sensitivity values. A ultra thin absorber made with combination of SRR is demonstrated [23] for imaging and

stealth applications. The size of this structure is very thin and found six resonance peaks. For sensing purposes, a practically perfect MMA was presented in [12]. At a frequency of 2.249 THz, they discovered a maximum absorption value of 99%. To verify their sensor performance for biomedical applications, they used refractive index values ranging from 1.3 to 1.39. The study's average sensitivity was 300 GHz/RIU and 23.7 GHz/RIU for both peaks. For use in microwave sensing applications, an Archimedes spiral-shaped metamaterial (MTM) absorber was developed [24]. With a high quality factor of 84.5 and a maximum absorption of 99.9%, the absorber displays seven resonance peaks. The proposal is for a triple-band mtm structure for terahertz applications [25]. For the three peak frequencies, this arrangement exhibits remarkable absorption values of 99%, 99%, and 100%, with peak sensitivities of 2.76 THz $\mu$m$^{-1}$ for analyte thickness and 1.55 THz RIU$^{-1}$ for refractive index detection. For RI index sensing applications, an ultra-narrow perfect mtm absorber in the thz region was proposed [26]. The study's greatest q-factor and absorption were 637 and 99.49%, respectively, and its sensitivity value was 506 RIU$^{-1}$. For the aim of sensing cooking oils, a metamaterial structure based on S.S.R.R. is shown [27]. At 3.04 and 3.44 GHz, the absorption values are 99.9 and 99.7, respectively, and the sensitivity values are 1410.29 and 1148.16.

In these studies, many researchers have proposed various metamaterial-based structures and sensors for biomedical applications. However, most of these designs face certain limitations. Some operate only in a single frequency band, while others demonstrate multiple absorption peaks but utilize only one for sensing. In many cases, placing a sensing layer alters the resonance characteristics or causes unstable performance when the refractive index changes. In contrast, a multi-band metamaterial absorber (MMA) enhances sensing reliability and reduces measurement error [28] by providing a consistent response across multiple distinct resonance modes. To overcome these issues, our study presents a novel, triple-band metamaterial absorber (MMA) with a compact footprint of $41 \times 41$ $\mu$m$^2$, which is designed without altering the commonly used substrate and materials. This development demonstrates creative structural innovation through precise geometrical tuning and multilayered symmetry that enables high absorption rates (above 99%) at all three resonant frequencies. A comprehensive parametric study was conducted to optimize the structure for maximum performance. Furthermore, E-field, H-field, and surface current distributions were analysed. A sensing layer was introduced to evaluate the sensor's refractive index sensitivity across all three peaks, which remained stable and showed high sensitivity. We further demonstrated the sensor's practical utility by detecting biological samples, including glucose, malaria-infected cells, and cervical cancer cells. Moreover, the sensor's suitability for microwave imaging (MWI) applications was also analysed. Overall, the sensor's high absorption, structural stability, and intense sensitivity make it a promising candidate for biomedical applications.

## Materials and methods

For the design and simulation of the structure, we use CST studio software, and the front and side views of this structure are depicted in Fig 1. Here, gold is used on the front and back sides, which have electrical and thermal conductivities of 4.561 $\times 10^7$ Sm$^{-1}$ and 314 W/K/m respectively. Gold possesses several physical properties, including a density of 19,320 kg/m$^3$, a specific heat capacity (Cs) of 0.13 J/K/Kg, a Young's modulus of 78 GPa, and a Poisson's ratio of 0.42 [17]. The substrate PTFE is used between these gold layer which has a electrical conductivity of 2.1. PTFE has outstanding thermal resistance and notable flexibility, and its low dielectric permittivity and loss tangent make it well-suited for THz sensing applications [29]. The size of the design is $41 \times 41$ $\mu$m$^2$ and the height is t = 2.1 $\mu$m. We use the cylinder option in simulation software to develop an octagonal shape, which has an outer radius and an inner

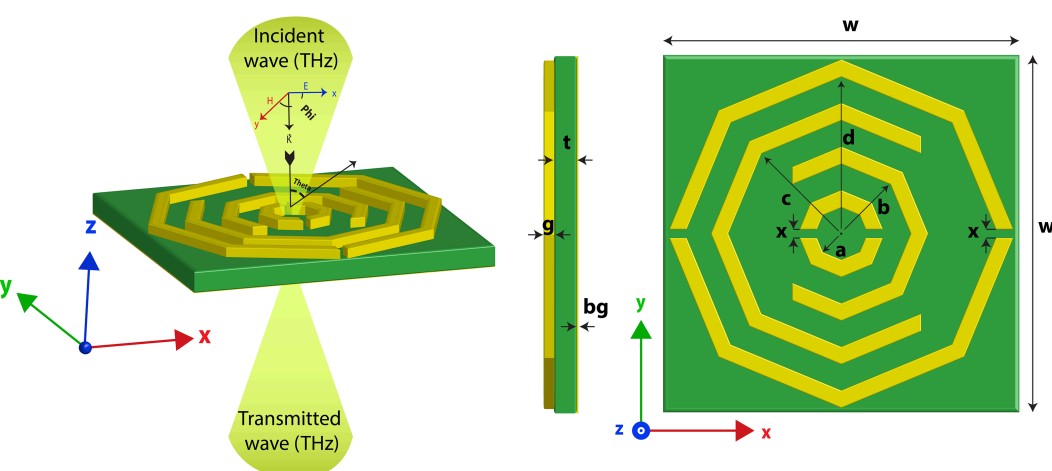

**Fig 1. Proposed MMA's simulation setup and structural pattern.**

radius option that helps to design more precisely. For the first octagon d, the outer radius is 20 $\mu$m, the inner radius is 18 $\mu$m and both sides are cut to a size of 1 $\mu$m. The same process was used to construct other octagonal structures and in the second octagon, we cut the right side to a size of 22 $\mu$m and left side of the third hexagon to a size of 15 $\mu$m. The last octagon is 5 $\mu$m and 3 $\mu$m size in terms of the in outer and inner radius and both sides are cut to a size of 1 $\mu$m. The background of this structure is built with a gold layer with a thickness of 0.18 $\mu$m, which prevents reflection and helps to achieve maximum absorption for this MMA. The height of the front part ,which is also made with gold, is 1 $\mu$m. The detailed parameter values for all the structures are shown in Table 1. Through an analysis of prior research on comparable metamaterial absorber designs, initial values for characteristics such as substrate width, height, octagonal width, and gap were determined. In addition, a thorough parametric analysis was carried out to optimise important parameters as needed to achieve high absorption efficiency and improved sensing performance. For the simulation, we use a Floquet port with different boundary conditions. For the x and y axes, the unit cell boundary condition is applied, and an open boundary condition is applied to the z-axis, the structure with directions shown in Fig 1. In the upper-left part of this figure, the polarization angle ($\phi$) and the incident angle ($\theta$) of the incoming wave are both set to 0 degrees, meaning the plane wave is normally incident along the z-axis with the electric field (E) polarized along the x-axis and the magnetic field (H) oriented along the y-axis.

Fig 2 represents the step-by-step design procedure and the absorption value. In the first step, we designed an octagon and cut both the right and left sides of it. For this structure, the single peak graph achieved a high absorption value. After that, a second octagon was added,

**Table 1. Parameter values for this suggested MMA.**

| Parameters | Sizes ($\mu$m) | | Parameters | Sizes ($\mu$m) |
|---|---|---|---|---|
| | Outer radius | Inner radius | w | 41 |
| d | 20 | 18 | t | 2.1 |
| c | 15 | 13 | bg | 0.18 |
| b | 10 | 8 | g | 1 |
| a | 5 | 3 | x | 1 |

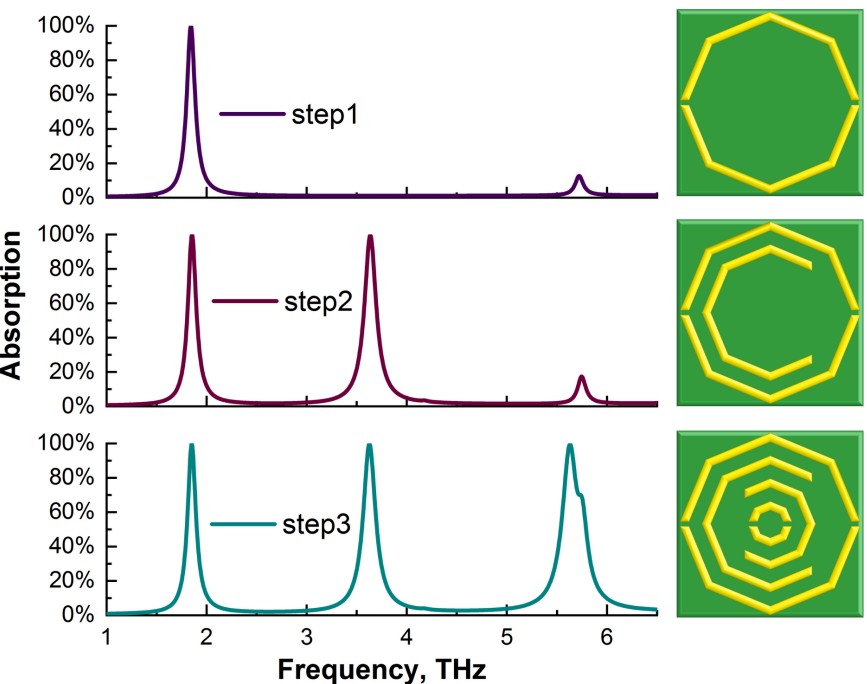

**Fig 2. Step-by-step MMA design procedure with absorption values.**

and the right side was cut, resulting in two peaks with high absorption values. Finally, another octagonal shape is added and cut to the left side, and inside of this shape, another octagon is added and cut on both sides to a size of one $\mu$m size. The last graph shows three peaks for this final design, all of which have high absorption values. Terahertz (THz) waves, spanning 0.1–10 THz, combine properties of both millimeter and infrared waves. Their non-ionizing nature, high resolution, deep penetration, and spectral fingerprinting make them promising for biomedical applications [30–32]. Our main aim was to design a multiband absorber with high absorption value in the THz region for biomedical sensing applications. Multipeak MMA sensors can help work with multiple modes of interest and also reduce error because they generate more sensing data, thus improving the reliability of the data [28]. This octagonal structure is ideal for achieving our goal by providing three distinct peaks in the region 1-6.5 THz with excellent absorption values. Therefore, we use only an octagonal shape in this structure.

Because this is a thin film structure, fabricating the absorber presents several challenges. Strict tolerances must be maintained in order to achieve accurate control over micro- and nanoscale dimensions. It may also be challenging to coordinate multiple layers within a complex design accurately. Material losses can also be caused by issues such as dielectric loss during processing, complex mask creation, and material incompatibility. The manufacturing outcome may also be influenced by environmental conditions, such as humidity and temperature [13,33–35]. Therefore, to effectively implement this concept in practice, sophisticated fabrication processes are needed. Additionally, this structure is straightforward, and we follow a ratio and pattern during our design, which makes it more helpful in the fabrication process. Many deposition methods can utilize this structure, such as chemical vapor deposition, sputtering, pulsed laser deposition and inkjet printing. For the sputtering method, first, a layer of PTFE is developed, and then, with the help of magnetron sputtering, a back and front gold

layer is deposited. Techniques such as photolithography and etching can help build the front structure [36]. Inkjet printing can then be used by depositing the gold structure in both sizes and a layer using a later pattern [37].

### Ethics statement

This study is based solely on computational simulations. All biomedical-related data used were obtained from previously published research works. No human participants or personal data were directly involved; therefore, ethical approval and informed consent were not required.

## Result analysis

### Absorption analysis

Absorption is a crucial parameter, as high absorption in a metamaterial absorber ensures maximum interaction with the incident electromagnetic wave. The simulation of this construction is carried out via 3D simulation software. The following formula is used to determine the absorption value for this layout:

$$A(\omega) = 1 - R(\omega) - T(\omega) \tag{1}$$

Here, the absorbance is denoted by $A(\omega)$, the reflectance is denoted by $R(\omega) = |S_{11}|^2$, and the transmittance is denoted by by $T(\omega) = |S_{21}|^2$. Thus, Eq (1) can be expressed as follows:

$$A = 1 - |S_{11}|^2 - |S_{21}|^2 \tag{2}$$

Since there is no transmission for the rear gold layer in this construction, it helps us obtain the highest absorption value. Consequently, the value decreases to zero. To write Eq (2) in that manner,

$$A(\omega) = 1 - |S_{11}|^2 \tag{3}$$

We can use a new equation to explain the absorption occurrences:

$$A(\omega) = 1 - R(\omega) = \frac{Z_\omega - n_0}{Z_\omega + n_0} \tag{4}$$

Here, $Z_\omega$ is the wave impedance, $n_0$ represents the free space wave impedance. To obtain the best absorption value, the free space should match the wave impedance. As Eq (4) states, when the real part of $Z_\omega$ is one, and the imaginary part is approximately 0, maximum absorption is achieved. The equation for reflection can be written as $R = |\frac{Z-Z_0}{Z+Z_0}|^2$ [38], where $Z = \sqrt{\mu/\epsilon}$ is the medium and $Z_0 = \sqrt{\mu_0/\epsilon_0}$ is the surrounding medium. A perfect absorption achieves maximum absorption when its impedance matches that of free space. This matching minimizes reflection, leading to high absorption. When the impedance of a medium is equal to that of its surroundings, reflections are eliminated $R = 0$, allowing the wave to pass through seamlessly and the equation becomes $A(\omega) = 1$, which ensures the maximum absorption value. Conversely, when the impedance is greater than that of the surrounding medium, absorption increases because the wave energy transitions more smoothly into the material, reducing abrupt changes.

CST simulation studio is used to simulate the structure, which is also used for absorption analysis. Fig 3(a) shows the absorption values of three peaks at frequencies of 1.85 THz,

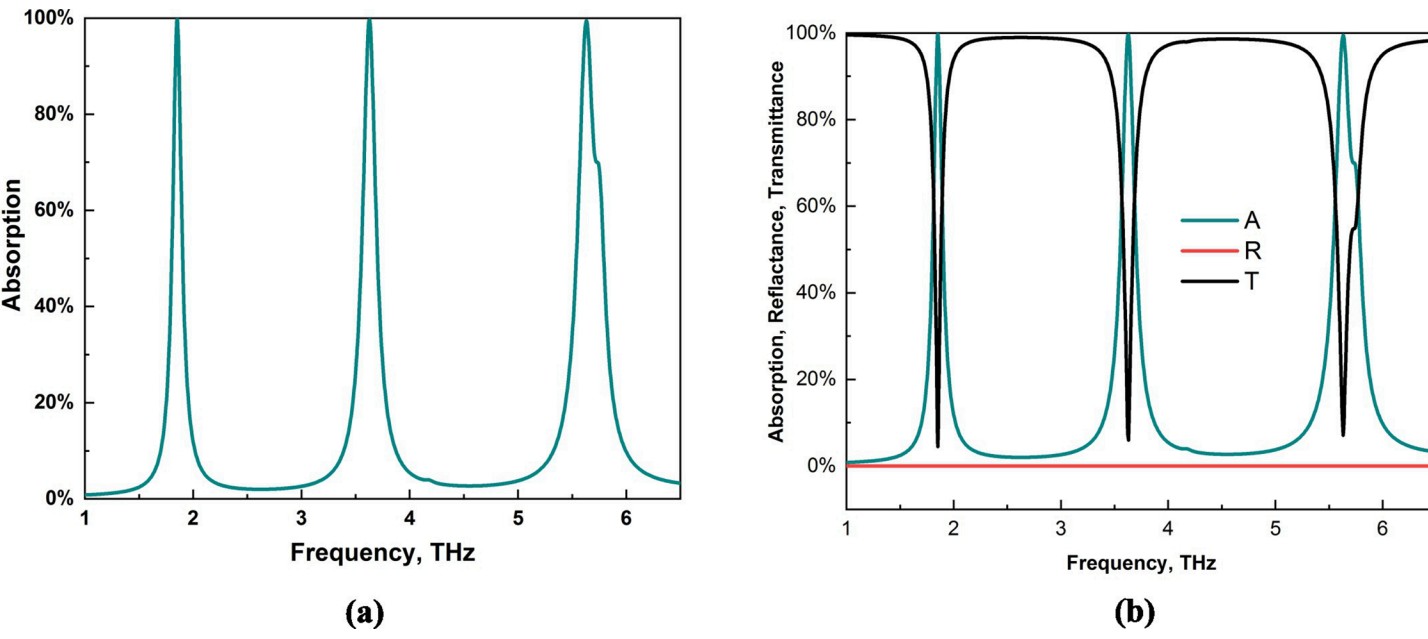

**Fig 3. (a) Absorption and (b) absorption, reflection and transmittance values of this MMA.**

3.62 THz and 5.63 THz, and the absorption values for these peaks are also high. For the first peak at 1.85 THz, the absorption value is 99.6%, which is the same as that for the second peak, where we also found 99.6%. Finally, the third peak at the 5.63 THz resonance frequency reached a 99.4% absorption value. This is a significant advantage for this layout because we identified three peaks, and the absorption value of each peak was above 99%. In Fig 3(b), the transmittance, reflection and absorption values are shown together. The transmittance value for the structure is zero due to the presence of the back gold layer used in this structure. This value also helps us achieve a high absorption value, as shown in Eqs (2) and (3).

Furthermore, we calculate the Q-factor value for these peaks because the quality factor is an important parameter for evaluating frequency mode performance. It also helps to identify how this device is effective in sensing applications [19]. This, in turn, helps to evaluate the overall effectiveness of the device in sensing performance. The equation for calculating the q-factor is where $f_0$ is the resonance frequency, and the *FWHM* or full-width half maximum is the value that is found by calculating the difference in 50% of the peaks. We used the data analysis tool Origin Pro to calculate the fwhm values and found values of 0.09673 for 1.85 THz, 0.14389 for 3.62 THz and 0.26787 for 5.63 THz. After calculating the q-factor for these frequencies, we determined that they are 19.13, 25.16, and 39.13. The highest value is in the third peak, and thus, it performs better for sensing applications, which we will evaluate later. Table 2 shows the performance analysis table for this MMA. Table 3 presents a comparison analysis with previous research. There are some triple-band absorbers that have been previously studied, where it can clearly be seen that maintaining all the peaks above 99% is difficult. However, this absorber works very well and maintains very high absorption values.

**Table 2. Performance analysis for this structure.**

| Resonance Frequency (THz) | Absorption (%) | FWHM | Quality Factor |
|---|---|---|---|
| 1.85 | 99.6 | 0.09673 | 19.13 |
| 3.62 | 99.6 | 0.14389 | 25.16 |
| 5.63 | 99.4 | 0.26787 | 39.13 |

**Table 3. Comparison of absorption with previous research.**

| Ref. | Resonance Frequency (THz) | Structure Size ($\mu$m) | Absorption (%) | Year |
|---|---|---|---|---|
| [39] | 3.5-4.1, 7.15 | 35 × 35 | 90, 98.9 | 2019 |
| [40] | 4.59, 6.18 | 2.4 ×2.4 | 99.91, 99.31 | 2019 |
| [41] | 1-3 | - | 97, 98, 99 | 2021 |
| [42] | 2-6 | 4.75 ×4.75 | 98.75 | 2021 |
| [43] | 0-3 | 80 ×80 | 99, 80, 95 | 2022 |
| [44] | 0.66–1.84 | 114 × 114 | 94.4 | 2023 |
| [45] | 2.638, 5.158 | 30 × 30 | Almost 90 | 2024 |
| Proposed | 1.85, 3.62, 5.63 | 41 × 41 | 99.6, 99.6, 99.4 | 2025 |

## Parametric study analysis

We utilize various types of parametric studies to identify the most suitable framework for our investigation. The absorption values for various substrate widths are displayed in Fig 4(a). The absorption value is high for a substrate size of 1.9 $\mu$m; however, it is slightly lower than that of 2 $\mu$m. Moreover, 2.2 $\mu$m and 2.3 $\mu$m have similar values, but 2.1 $\mu$m is the ideal value for this structure because it helps in determining the overall best outcomes. We analyze different widths for the front gold structure shown in Fig 4(b). Our findings indicate that the absorption values remain nearly consistent across these various sizes. This information is essential for evaluating the performance of the gold layer, specifically in configurations measuring 1 $\mu$m thick. To analyze the optimal performance, we also altered the substrate materials presented in Fig 4(c). Polycarbonate and polyimide have relatively low absorption results. Two of the peaks for these materials fall below 90%, whereas the other two are not in the best positions. The results are better for Rogers RT5880, and every peak reaches 99%. However, PTFE yields superior results than this, so we employ it for this construction. To determine the best materials for our layouts and improve the sensing performance, we simulate all of these materials, which are among the most commonly used materials.

## Equivalent circuit analysis

As illustrated in Fig 5(a), we developed an equivalent circuit via Advanced Design System (ADS) software, producing results that align closely with those obtained from CST, as shown in Fig 5(b). To construct this circuit, we replaced microstrip lines with inductors and utilized capacitors to represent the split gaps. Specifically, capacitor C1 and inductor L1 contribute to the first peak frequency of 1.8 THz. Similarly, capacitor C2 and inductor L2 correspond to the second peak frequency, which is 3.6 THz. For the third peak frequency, which occurs at 5.6 THz, we incorporate capacitors C3 and C4 along with inductors L3 and L4. Moreover, we included capacitors C5, C6, C7, and C9 to replace the gaps between the octagonal rings, accurately representing the spacing between them. The results indicate that the equivalent circuit closely resembles the unit cell and provides an output that aligns with the results obtained from CST. Notably, while the CST results are static, tuning is required in the ADS results,

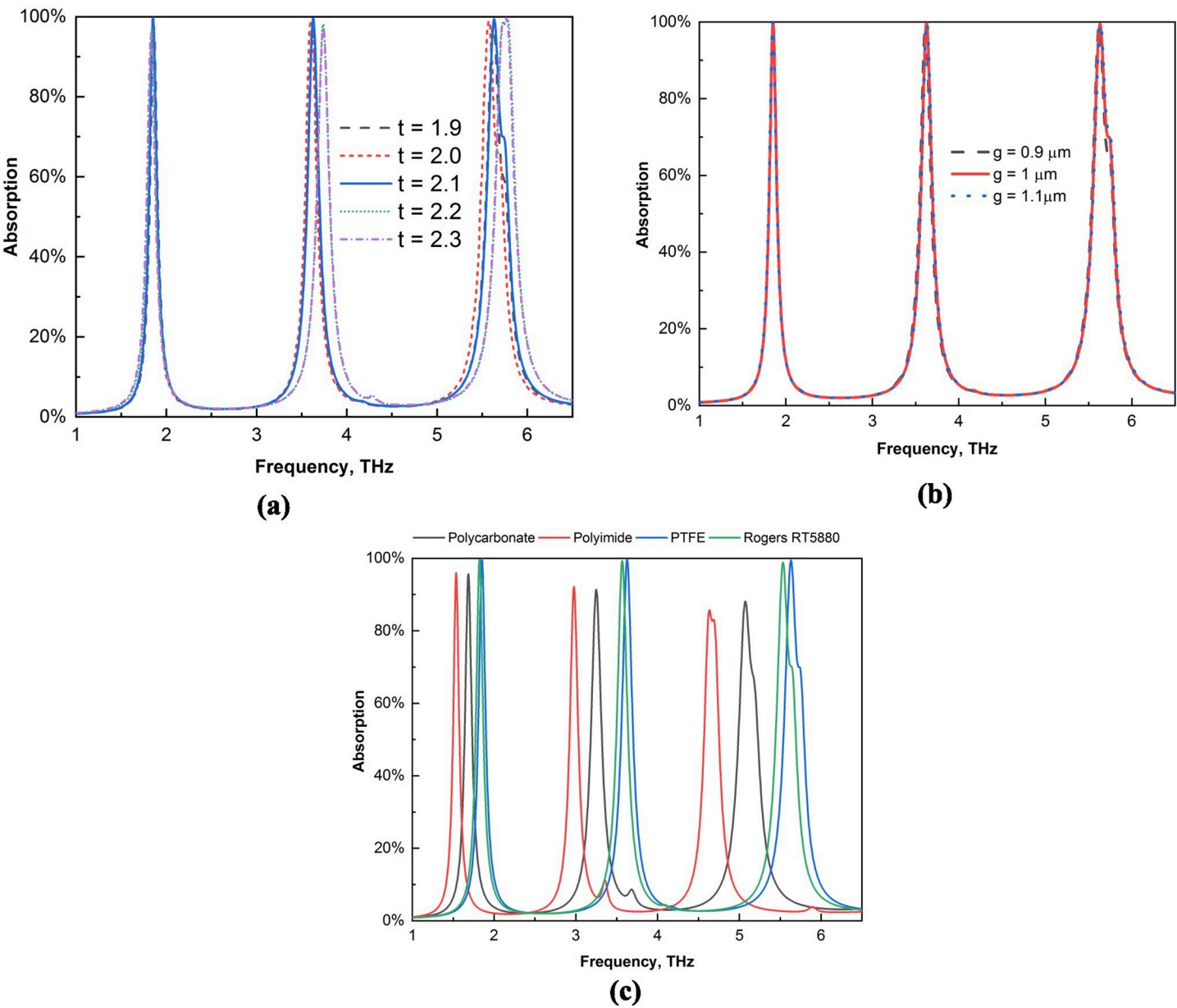

**Fig 4. Parametric study for (a) different substrate widths, (b) front gold structure widths, and (c) different substrates.**

which accounts for the observed differences. Nevertheless, the peak frequencies remain in the same positions across both methodologies.

## Angle stability

As shown in Fig 6, this study also examines the absorption value for various incidence and polarization angles. This structure is incident up to a 45-degree polarization angle, as shown in Fig 6(a). Although there is a slight peak at 45°, the absorption values for the three resonance frequencies are essentially the same. The first and third peaks shift at 60°, whereas the second peak remains the same. Therefore, this structure, especially our second peak, can be

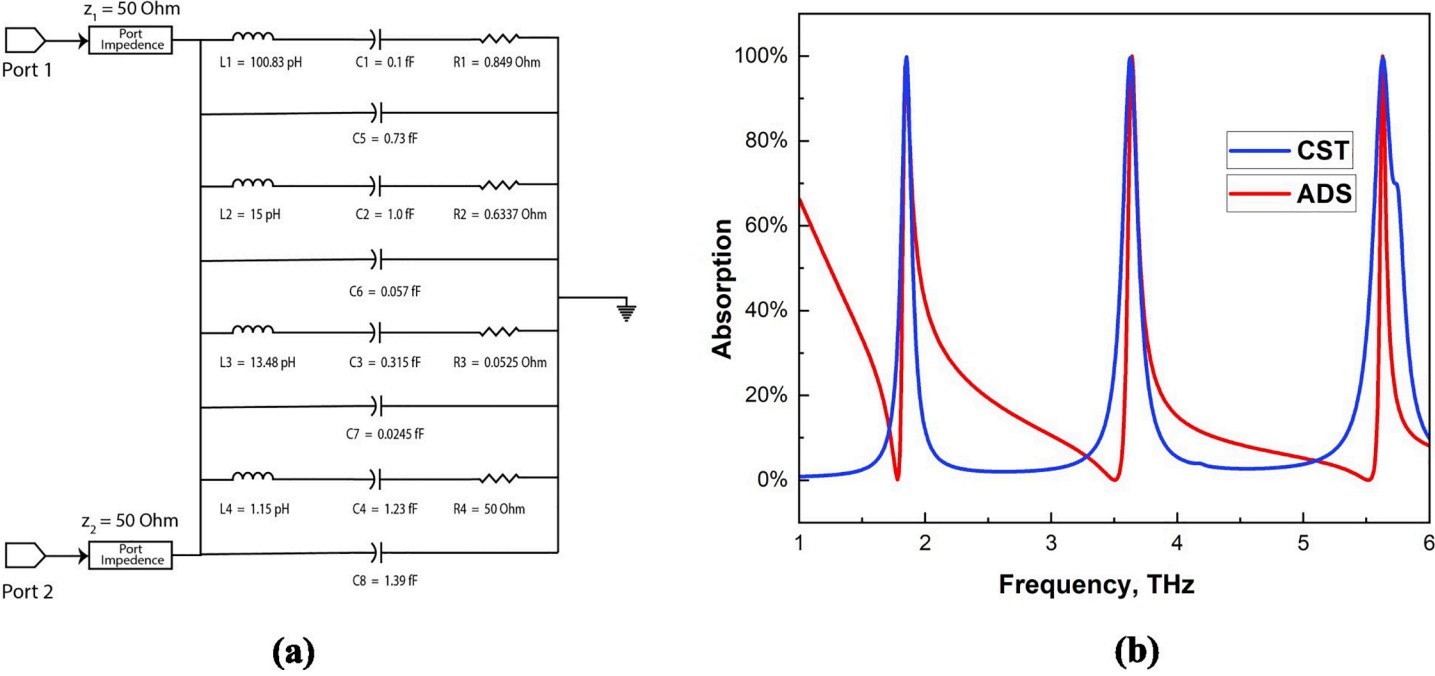

**Fig 5. Suggested structure: (a) equivalent circuit and (b) absorption result of the ADS and CST comparison.**

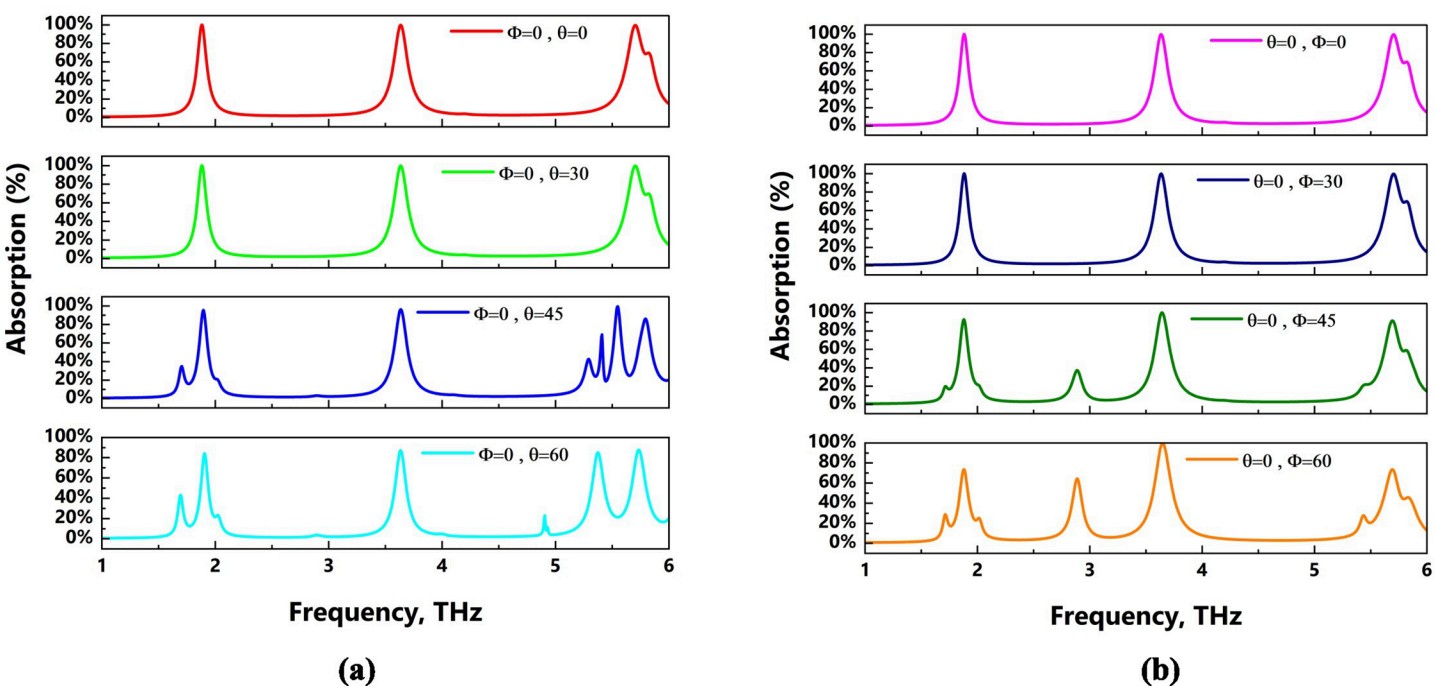

**Fig 6. Absorption values for (a) different polarization angles and (b) different incident angles.**

used at larger polarization angles. The same phenomenon occurs at the peak at 3.62 THz for the various incidence angles presented in Fig 6(b). All of the peaks, however, remain until 30°, after which the first and second peaks begin to distort, but we still detected a number of peaks with significant absorption values. These angle shifts are caused by impedance mismatching conditions [46], which result from variations in the interaction position of incident light within the structure. These conditions cause polarisation mismatch and variations in optical path length, particularly at oblique angles like 60°, where these distortions become more noticeable. However, for this kind of incidence angle, we can utilise these peaks if needed, as we also observe many absorption peaks here.

## E-Field, H-field and surface current distribution

The electric field, magnetic field, and surface current distribution of the proposed MMA are interrelated and can be elucidated via Maxwell's equations, as described below [47],

$$\nabla \times H = J + \epsilon \frac{\delta E}{\delta t} \tag{5}$$

$$J = \sigma E \tag{6}$$

The E-field distribution for the 1.85 THz, 3.62 THz, and 5.63 THz frequencies are shown in Fig 7. The phenomenon known as surface plasmon resonance [48] is the reason why the e-field strength appears as a red patch. Owing to cavity surface plasmon resonance (CSPR) [49], the split gap at the top and bottom exhibits a strong electric field at a frequency of 1.85 THz. Additionally, localised surface plasmon resonance (LSPR) produces an e field on the left and right sides of the octagonal form. Moreover, on the left and right sides of the octagonal shape, an e field occurs due to localised surface plasmon resonance (LSPR) [50], which produces an e-field on the left and right sides of the octagonal form. CSPR and LSPR cause a second spiral from the inside and a strong electric field in the split gap at a frequency of 3.62. Finally, for the last peak frequency, a strong e field can be observed at the second octagonal edge and top side because of the LSPR.

The distribution of the magnetic field for each frequency is shown in Fig 8. Because of the LSPR and CSPR, a strong magnetic field is present on the right side of the structure and in the center of its left side at lower frequencies. The structure's left and right sides exhibit a significant h field concentration at a frequency of 3.62 as a result of LSPR. In the middle of this structure, CSPR occurs because of the strength area in the middle octagon, and the second octagon from the middle is an intense red area with the last frequency. An LSPR phenomenon is observed for the third octagon.

Additionally, as illustrated in Fig 9, this study analyzes the surface current distributions for three different frequencies. The upper side of the structure exhibits an intense concentration of surface current at a frequency of 1.85 THz. A strong magnetic field is indicated by the presence of antiparallel current flows [51]. Owing to the concentration of parallel and antiparallel current flows, the upper side of the first two octagons shown in Fig 9(a) generates a high magnetic field. Next, the second octagonal shape from the top produces a strong antiparallel current flow and good current flow for 3.62 THz seconds, which results in a strong magnetic field in this region. A surface current of 5.63 THz is shown in Fig 9(c), where a strong magnetic field is present in the second octagonal shape from the middle, and some antiparallel current is present on the lower side of the central structure.

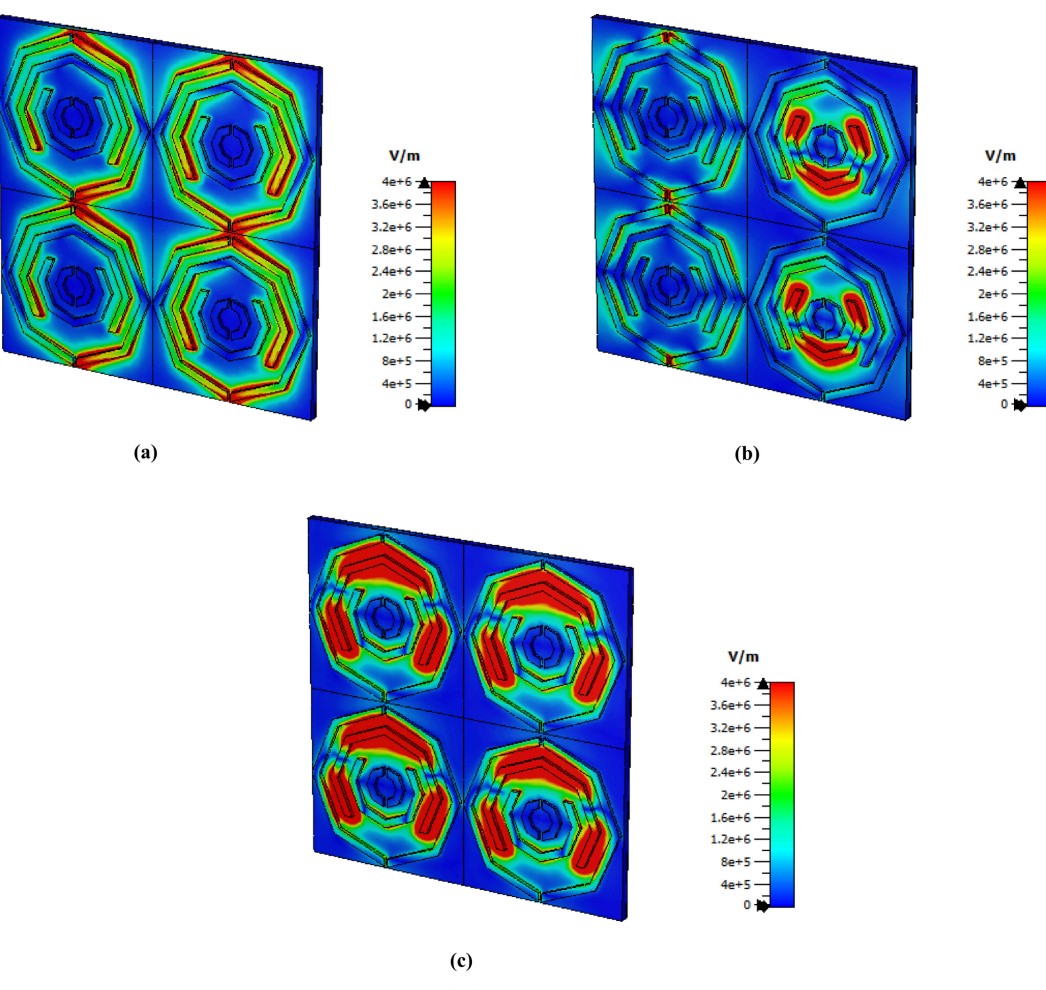

**Fig 7. E field distribution at (a) 1.85 THz, (b) 3.62 THz and (c) 5.63 THz.**

## Sensing performance analysis

Fig 10(c) shows the sensor configuration, which includes a 3 $\mu$m sensor that can be used for sensing applications. The sensor layer is placed above the substrate, and the top gold resonator layer lies within this configuration, initially left vacant to allow for sample insertion. To introduce the test object, a liquid form of the analyte is dropped or injected into the sensor layer from the top side, allowing it to fill the sensing region. This setup ensures direct interaction between the test sample and the resonator, enabling changes in the dielectric environment to be detected through measurable shifts in the resonance frequency. The experimental setup for this sensor is shown in Fig 10(a). where a THz source crosses the sensor with a frequency range of 1-6 THz. When these data are further processed and detected, we can observe different types of absorption graphs for these samples. A bottom view of this sensor is shown in Fig 10(b) to view the sensor construction more clearly.

We must examine certain crucial electromagnetic characteristics, such as effective permittivity ($\epsilon_{eff}$), effective permeability ($\mu_{eff}$), refractive index ($n$), and wave impedance ($z$), to understand the metamaterial's behaviour. These variables are essential to understanding the sensor layer. They can be computed using several well-established methods, including the

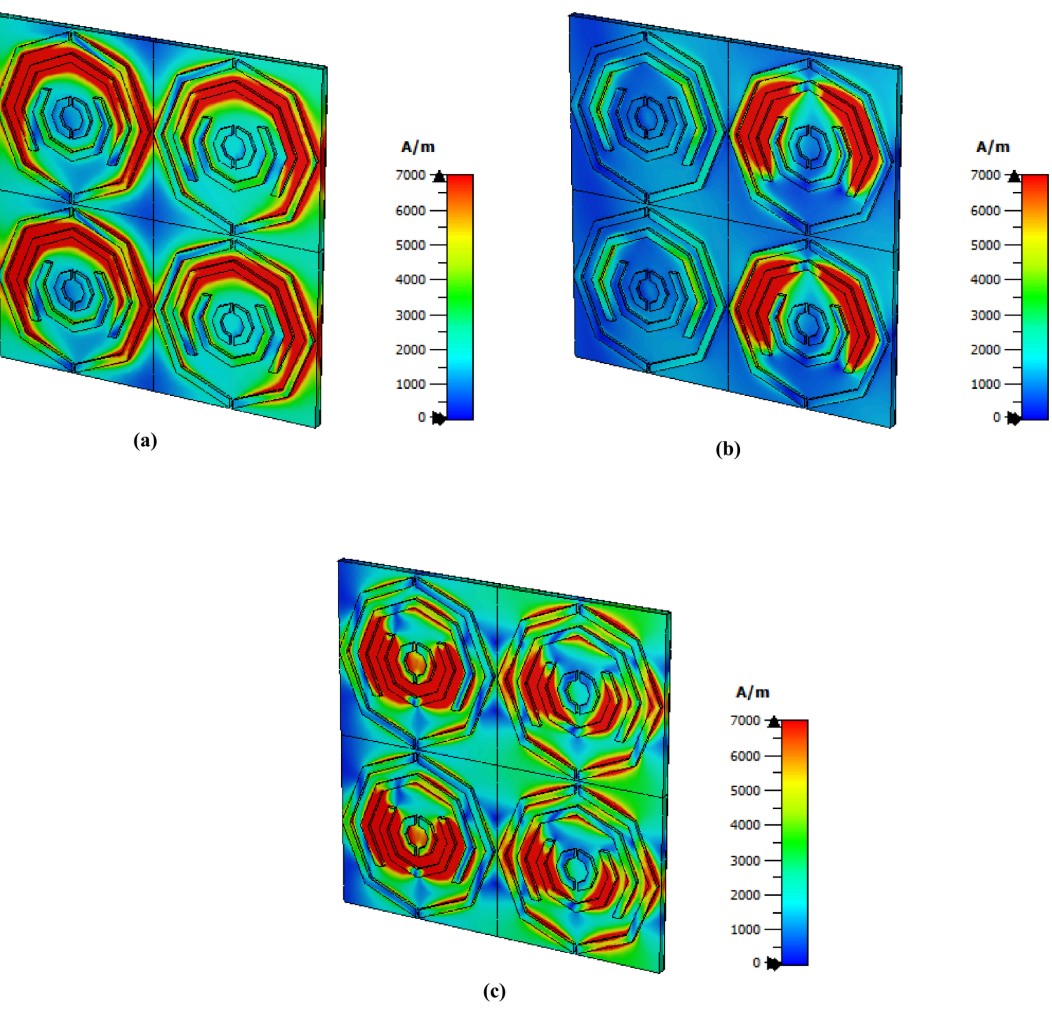

**Fig 8. H field distribution at (a) 1.85 THz, (b) 3.62 THz and (c) 5.63 THz.**

Lossy-Drude models, the Nicolson-Ross-Weir method, and the transmission-reflection (TR) method. The following formula is used to determine n in both the Nicolson-Ross-Weir and Lossy-Drude methods: $n = \cos^{-1}\left((1/k\delta) \cdot \left((1 - S_{11} + S_{22})/(2S_{21})\right)\right)$. The wave impedance $z$ is obtained as follows: $z = \sqrt{\left((1 + S_{11})^2 - S_{22}\right)\big/\left((1 - S_{11})^2 - S_{22}\right)}$. Particularly when multiple sensing layers are present, these derived characteristics help describe the electromagnetic response of the structure [24].

Since the primary goal of this sensor is to be used in biomedical applications, we first examine its ability to detect various refractive indices. We first test its performance between 1.34 and 1.40 RI values, as the majority of biological samples have RI values between 1.3 and 1.39. Fig 11 illustrates how the sensor's absorption measurements change once these values are inserted. Fig 12 shows a close-up of each peak. For RI values of 1.34, 1.36, 1.38, and 1.4, we discovered peaks in our first band at 1.685 THz, 1.6718 THz, 1.6633 THz, and 1.6515 THz, respectively. For every peak, the absorption value is greater than 97%.

The frequency values for the second peaks are 3.27 THz, 3.25 THz, 3.23 THz, and 3.21 THz, and the absorptions for each peak are greater than 99%. The final peaks were located at

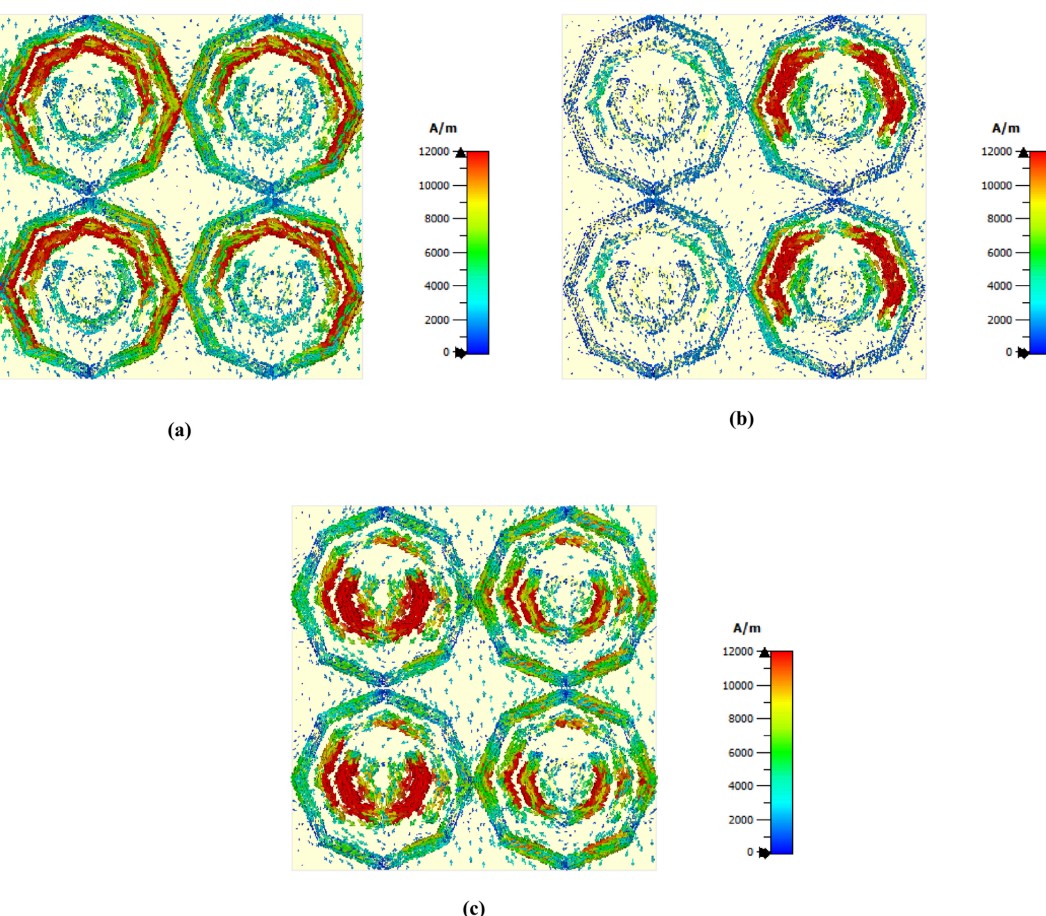

**Fig 9. Surface current field distribution at (a) 1.85 THz, (b) 3.62 THz and (c) 5.63 THz.**

5.155 THz, 5.125 THz, 5,0979 THz, and 5.07 THz, with corresponding RI values of 1.34, 1.36, 1.38, and 1.40, respectively. For various numbers, the absorption value is likewise greater than 99%. The fact that this sensor produces a high absorption value even after a sensor is added and that all the peaks remain the same but shift for various sensor values makes this an ideal choice for sensing applications.

Sensitivity is a key parameter that defines the sensor's ability to detect and differentiate minor variations in a sample's permittivity [52]. Higher sensitivity values at each resonance demonstrate the device's strong interaction with the analyte, enabling reliable detection performance. Eq (7), where $f_0$ is the shift in frequency peaks, and $n$ is the difference in the refractive index value, is used to compute the sensor's sensitivity.

$$S = \frac{\Delta f_0}{\Delta n} \tag{7}$$

The maximum sensitivity for this proposed sensor is an impressive 1500 GHz/RIU at the third peak, followed by 1000 GHz/RIU at the second peak and 660 GHz/RIU at the first peak. These remarkable values not only highlight the sensor's exceptional ability to detect even the slightest changes in the permittivity of various substances but also position it as a leading tool in the biomedical field. The sensitivity performance of this sensor is compared with that of

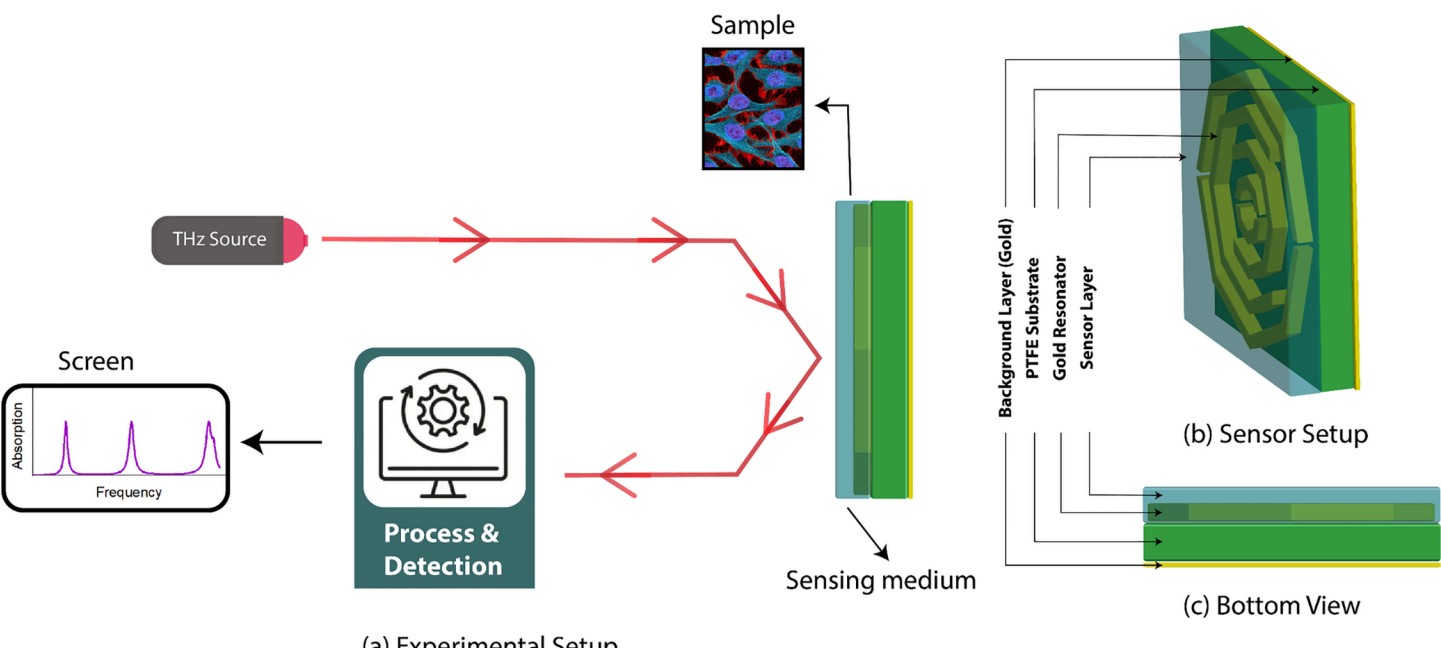

**Fig 10. (a) Experimental setup (b) Sensor setup (c) Bottom view with sensor.**

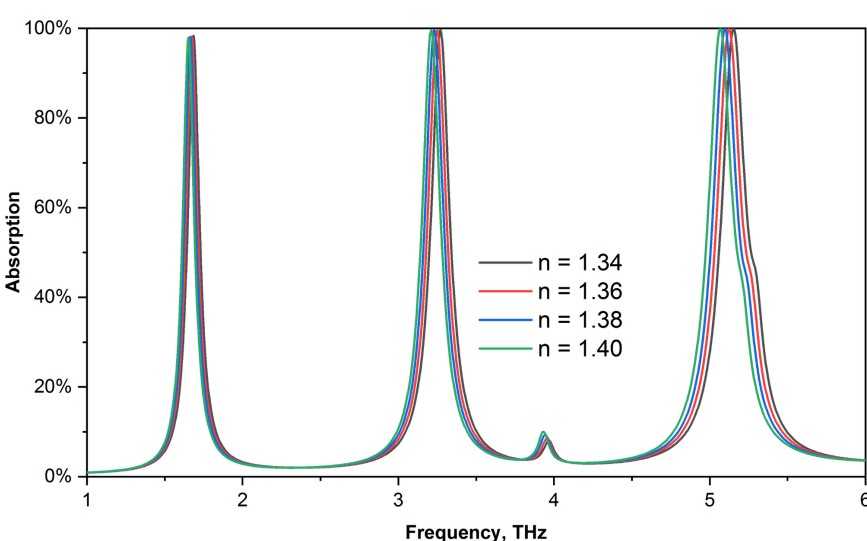

**Fig 11. Effect on absorption for different refractive index values.**

earlier studies in Table 4. There are various methods to boost this sensor's performance, which can be used to expand its possibilities. Better sensitivity could be achieved by altering the structure or by employing different materials and substrates within the existing arrangement. Its performance can also be improved by modifying the analyte layer's thickness or design parameters.

A sensor for the detection of microorganisms in the 0–4 frequency range was proposed by Bhati et al. [54]. The sensitivity values for all the peaks of this proposed sensor are higher than

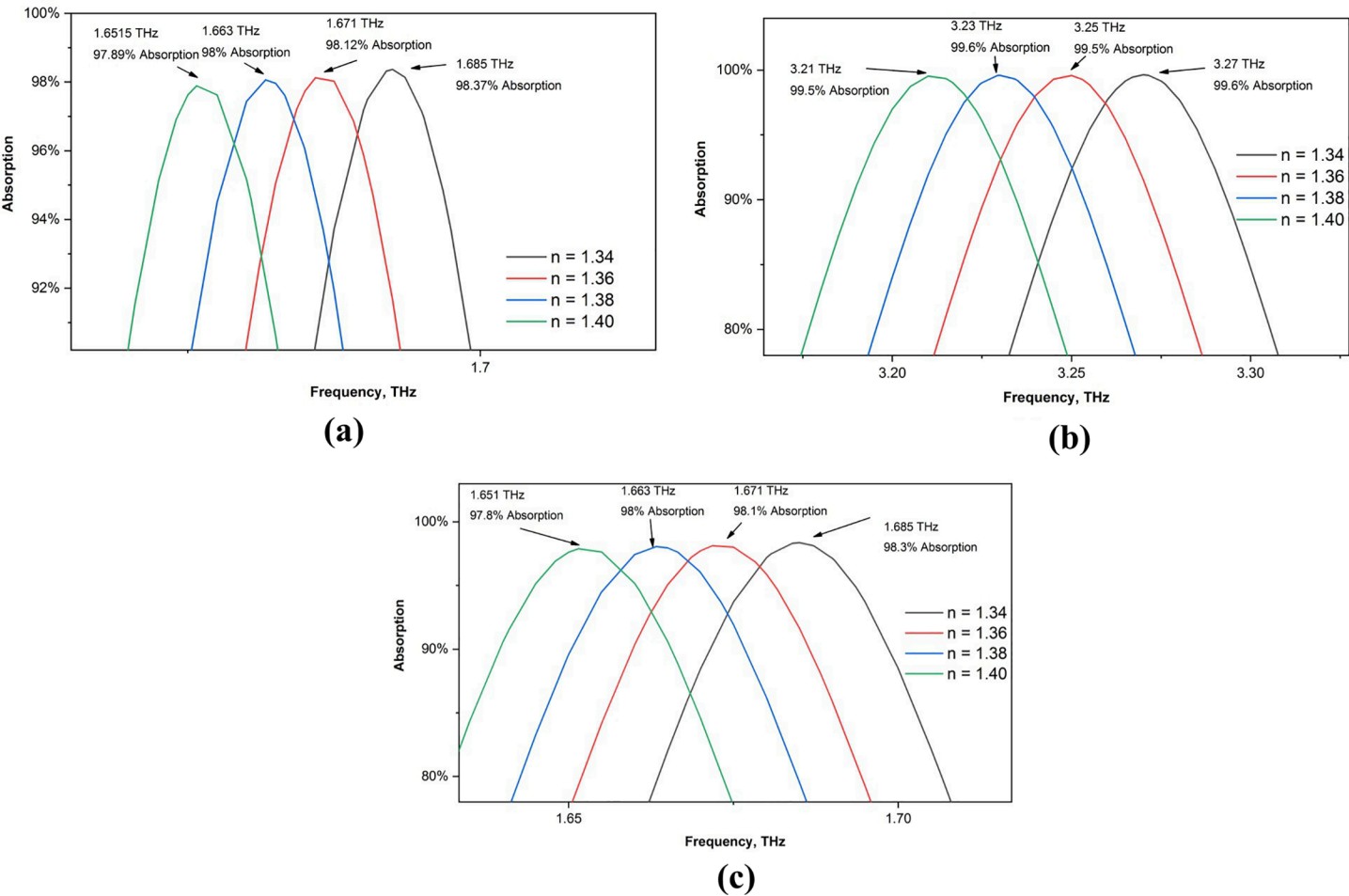

**Fig 12. Enlarged view of the absorption shift for different RI value (a) first peak (b) second peak (c) third peak.**

**Table 4. Comparison of Sensitivity performance with recent research.**

| Ref. | Operating Frequency (THz) | S THz/RIU | Year Published | Publication in | Applications |
|---|---|---|---|---|---|
| [53] | 1.4-2.8 | 0.13, 0.14 | 2024 | Photon. Nanostruct. Fundam. Appl. | sensing |
| [54] | 0-4 | 0.103, 0.095 | 2023 | Scientific Reports | Microorganisms detection |
| [55] | 1-1.8 | 0.278 | 2023 | Photonics | Bovin serum,albumin protein |
| [51] | 0-1.2 | 0.0515, 0.076 | 2023 | IEEE Access | Non-melanoma skin cancer detection |
| [56] | 1-1.6 | 0.281 | 2023 | Photonics | biosensor |
| [14] | - | 0.1057 | 2022 | IEEE sensors | Biomedical sensing |
| [57] | 0-1 | 0.0968, 0.1182 | 2022 | American chemical Society | Biomedical Application |
| [58] | 0.5-2.5 | 1.21 | 2022 | Nanomaterials | Breast cancer detection |
| [59] | 0.1-3 | 0.3, 0.912 | 2022 | Biosensors | Sensor |
| [42] | 2-6 | 0.851 | 2021 | Plasmonics | RI biosensor |
| This Work | 1-6 | 1.5, 1, 0.66 | 2025 | - | Biomedical Application |

the 0.103 THz/RIU and 0.095 THz/RIU sensitivities that they discovered. Subsequent studies by Singh et al. [14] and Hamza et al. [51] revealed values of 0.1057 Thz/RIU, 0.0515 THz/RIU and 0.076 THz/RIU, which are lower than those reported in this work. Nickpay et al.'s [42] refractive index biosensor research revealed a satisfactory sensitivity value of 0.851 THz/RIU in the 2–6 THz working range. The value is encouraging, and the second and third peaks outperform this suggested sensor. Therefore, it is evident from the comparison that this sensor performs exceptionally well and is ideal for use in biological applications. To further establish its performance in the biochemical field, this study additionally assessed the sensing performance for several biochemical sample types.

## Biochemical sensing

Glucose is a vital molecule for the human body, as it acts as a primary energy source for cells and supports essential metabolic functions [60]. As a result, we used this sensor to identify these crucial components. The refractive index of water is 1.3198, and the RI value of water containing 25% glucose is 1.3594 [61]. To identify any frequency change between these two numbers, we input these data into our sensor. There is a significant difference between these two, as shown in Fig 13. Additionally, Fig 14 displays a broader perspective of the absorption impact for the three peaks. There is a significant shift in these RI values for glucose and water. Our third peak had the most considerable frequency shift.

## Malaria detection

The detection of malaria is crucial due to its significant impact on human health. Healthy red blood cells (RBCs) present a refractive index value of approximately 1.373. In contrast, those infected with malaria demonstrate refractive indices of 1.373 during the schizont phase and 1.383 during the trophozoite phase [19]. The use of these refractive index (RI) values in the sensor leads to observable frequency shifts, as illustrated in Fig 15. To facilitate a deeper understanding of these frequency shifts, Fig 16 provides a magnified view of the three distinct frequencies, allowing for a more detailed analysis of the shifting values.

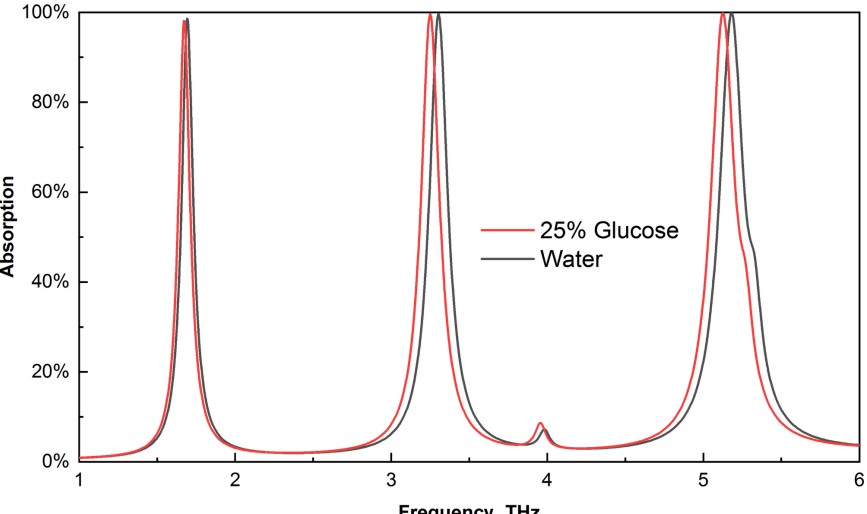

**Fig 13. Effects on the absorption of water and 25% glucose.**

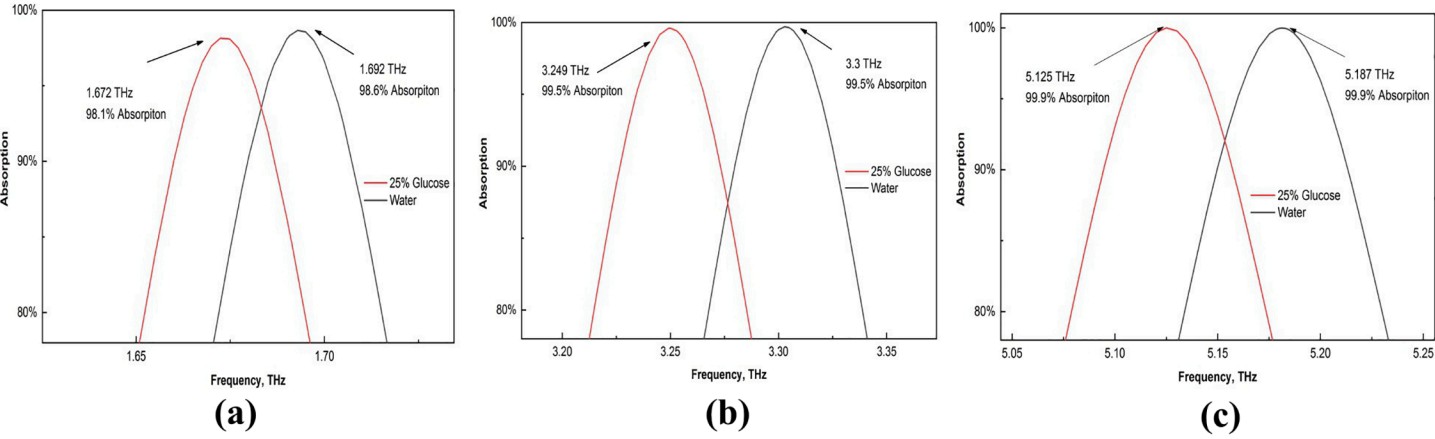

**Fig 14. Enlarged view of the absorption shifts for water and 25% glucose (a) first peak (b) second peak, and (c) third peak.**

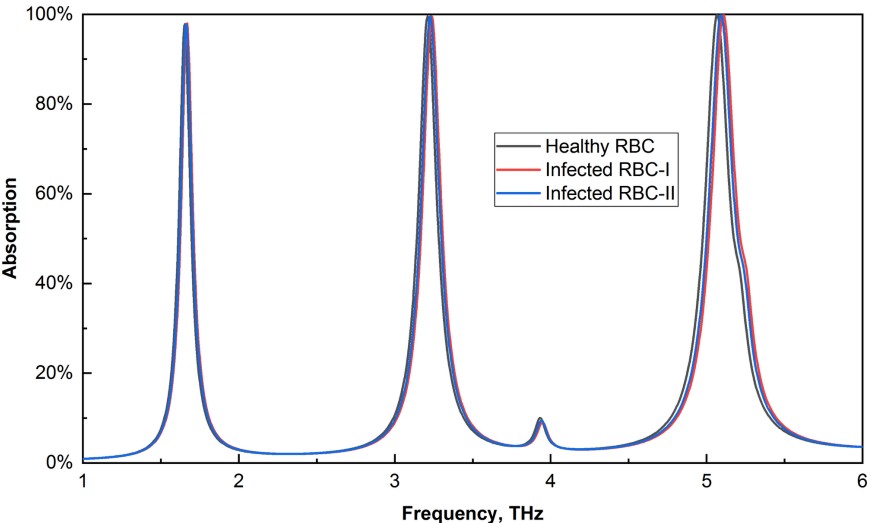

**Fig 15. Effects on the absorption of healthy and infected RBCs.**

## Cancer detection

Cervical cancer remains a significant health concern, and early detection can be achieved through the analysis of normal and malignant HeLa cells. The refractive indices of normal and cancerous HeLa cells are reported to be 1.368 and 1.392, respectively [62]. These values were utilized in our sensor to assess the frequency shift, revealing a substantial change in absorption associated with these indices, as demonstrated in Fig 17. Furthermore, we present an enlarged view of these findings in Fig 18 to facilitate a more precise examination. The sensor demonstrates exceptional performance in detecting cancerous cells, indicating its potential for use in clinical settings for cervical cancer diagnostics.

Different refractive indices at predetermined intervals, multiple fixed values for individual samples, and a mixed sample of glucose and water were used to assess the sensor's performance. The results of these studies demonstrate that the sensor performs effectively with both mixed and pure biological materials. Furthermore, it has the potential to be applied

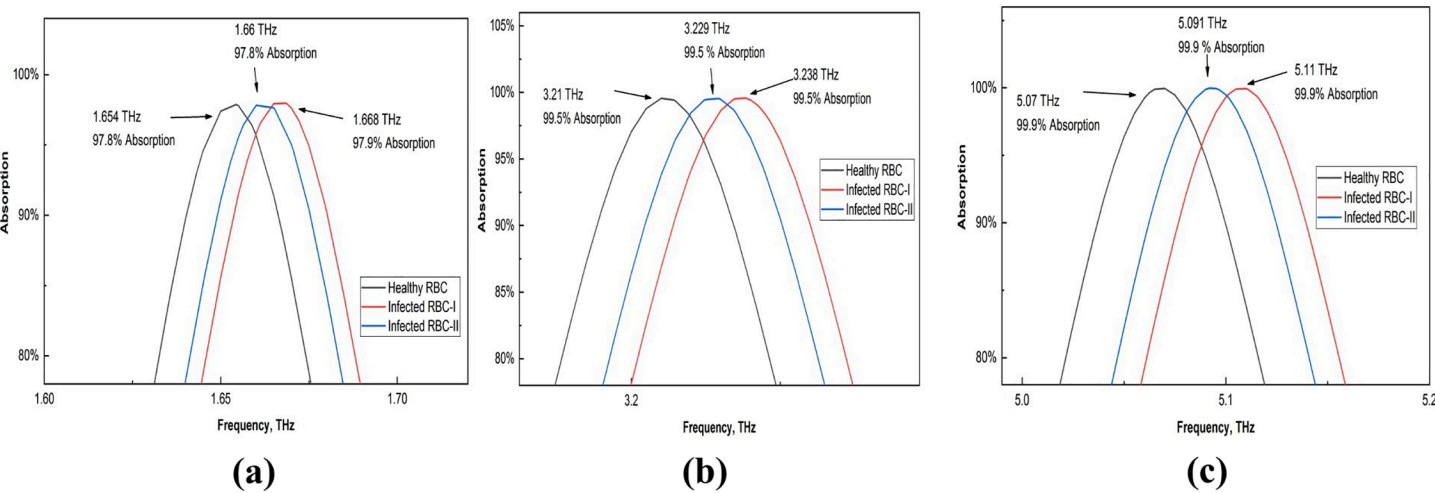

**Fig 16. Enlarged view of the absorption shift for healthy and infected RBCs (a) first peak (b) second peak, and (c) third peak.**

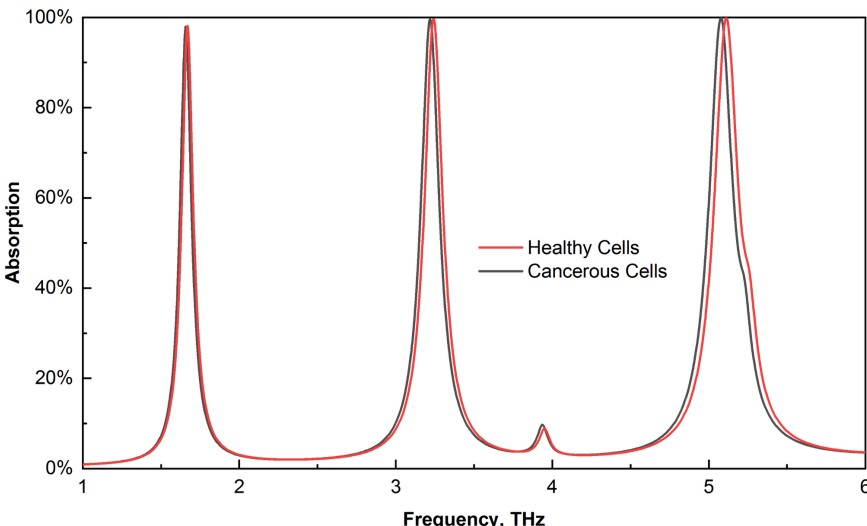

**Fig 17. Effects on the absorption of healthy and cancerous cells.**

across various biochemical contexts, thereby advancing technology while enhancing cost-effectiveness and reducing time consumption in research and diagnostics.

The use of E-file and H-field in microwave imaging (MWI) technology to identify cancerous and healthy cells has also been covered in several studies [16,63]. These techniques are also presented in this article for both malignant and normal HeLa cells. Cancerous cells exhibit superior intensity in electric and magnetic fields because they have a higher refractive index than normal cells do. The MWI approach for malignant and normal cells at 1.85 THz is displayed in Fig 19. Compared with normal cells, cancer cells have a redder field, especially on the upper side.

The cancer cells exhibit greater intensity than the normal cells do at 3.62 THz as shown in Fig 20. Finally, Fig 21 presents the electric field MWI result at 5.63 THz; although the intensity is not showing better than that of the other values, it can clearly be seen that there is a

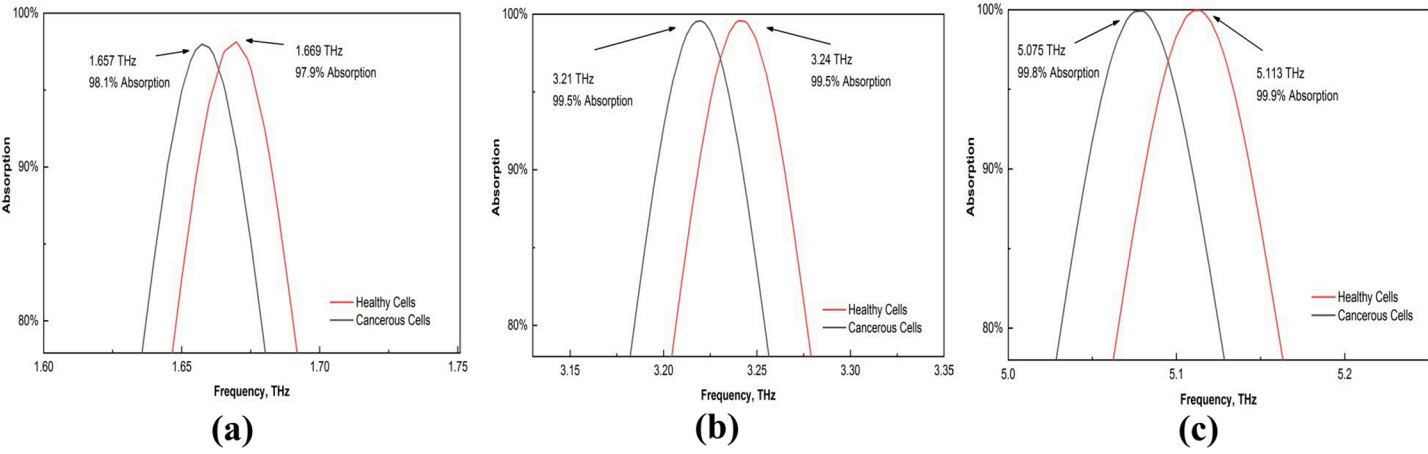

**Fig 18. Enlarged view of the absorption shift for healthy and cancerous cells (a) first peak (b) second peak, and (c) third peak.**

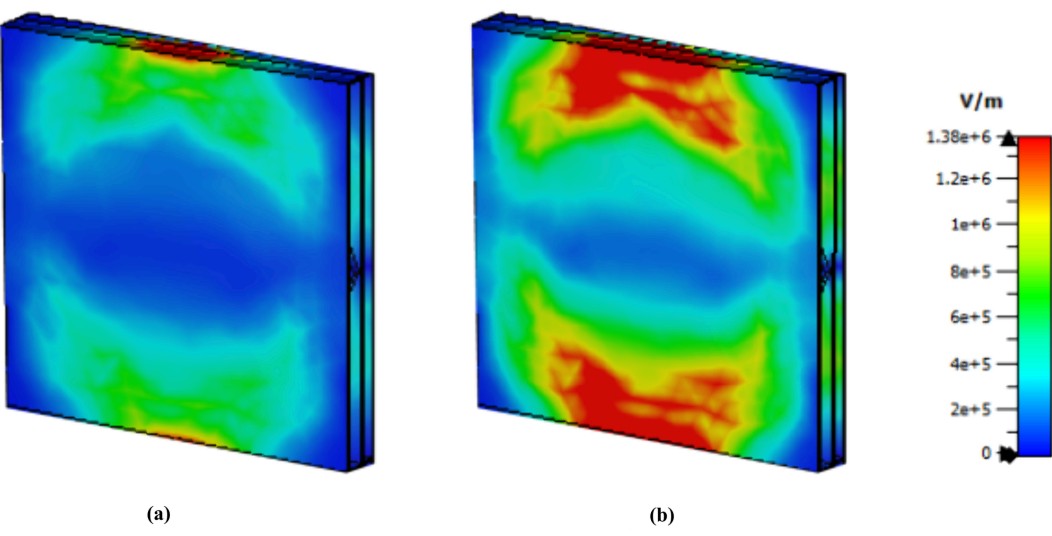

**Fig 19. E field MWI technique at 1.85 THz (a) normal cell (b) cancer cell.**

difference between these two values. Therefore, this sensor can be used to detect normal and cancer cells via the MWI technique.

The H-field Microwave Imaging (MWI) technique is a sophisticated tool for nondestructive evaluation, offering a unique ability to detect and visualize internal defects or anomalies within materials. By transmitting microwave signals and capturing the resulting magnetic field responses, MWI provides valuable insights into material properties and structural integrity. This method is particularly beneficial in industrial settings, such as manufacturing quality control and the inspection of composite materials, where precise defect detection is crucial. Additionally, its potential applications extend into medical imaging, where it could enhance diagnostic capabilities by providing detailed, noninvasive assessments of biological tissues [16].

The further development of the H-field MWI technique has been employed to effectively differentiate between healthy and cancerous cells as shown in Figs 22, 23 and 24. Both the

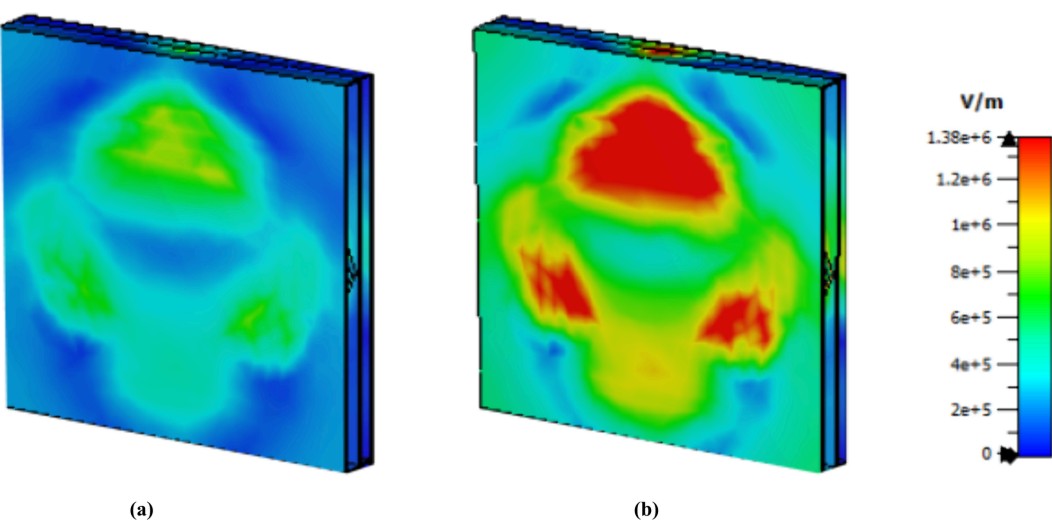

**Fig 20. E field MWI technique at 3.62 THz (a) normal cell (b) cancer cell.**

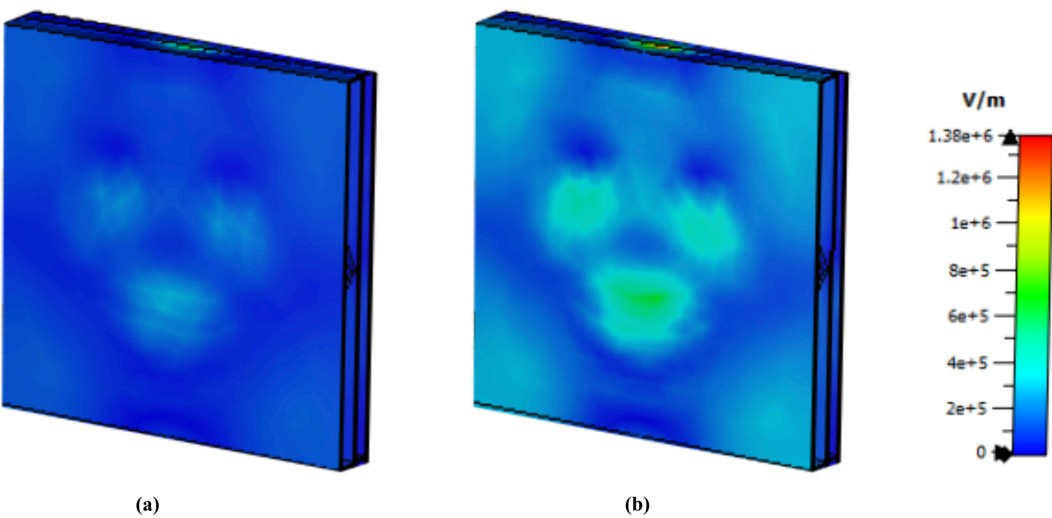

**Fig 21. E field MWI technique at 5.63 THz (a) normal cell (b) cancer cell.**

first and third peaks yield promising results; however, the second peak has a lower intensity value for both normal and cancer cells. This observation suggests that employing diverse sensing techniques, such as imaging along with graphical analysis methods, could increase the sensitivity and accuracy of this sensor in biochemical applications.

The figure of merit (FOM) is a crucial element for comparing the sensing capabilities of different sensors [13]. Our relatively high FOM values confirm that the proposed absorber performs competitively with or better than many existing THz sensors. Eq (8) [12] is used to calculate the FOM, where S and the FWHM are the sensitivity and full-width half maximum, respectively, which were calculated earlier.

$$FOM = \frac{S}{FWHM} \tag{8}$$

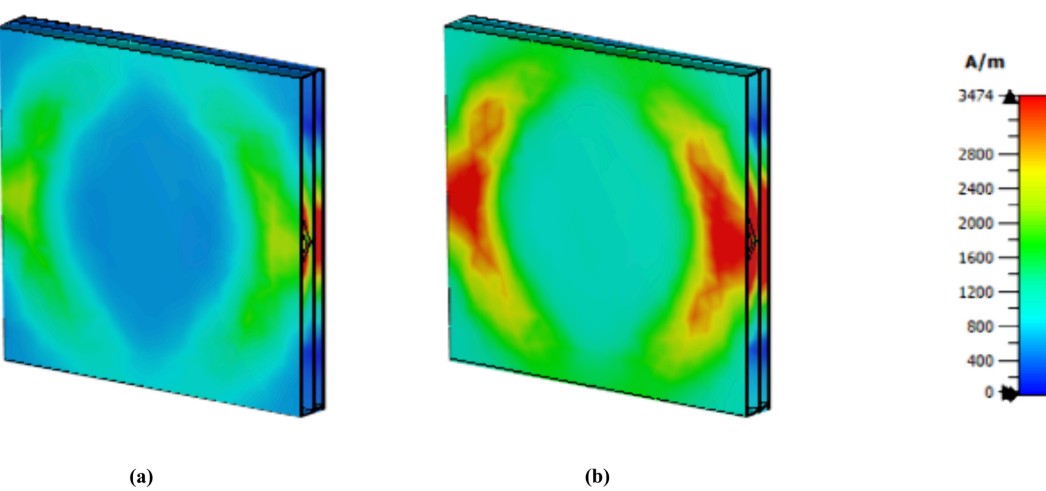

**Fig 22. H field MWI technique at 1.85 THz (a) normal cell (b) cancer cell.**

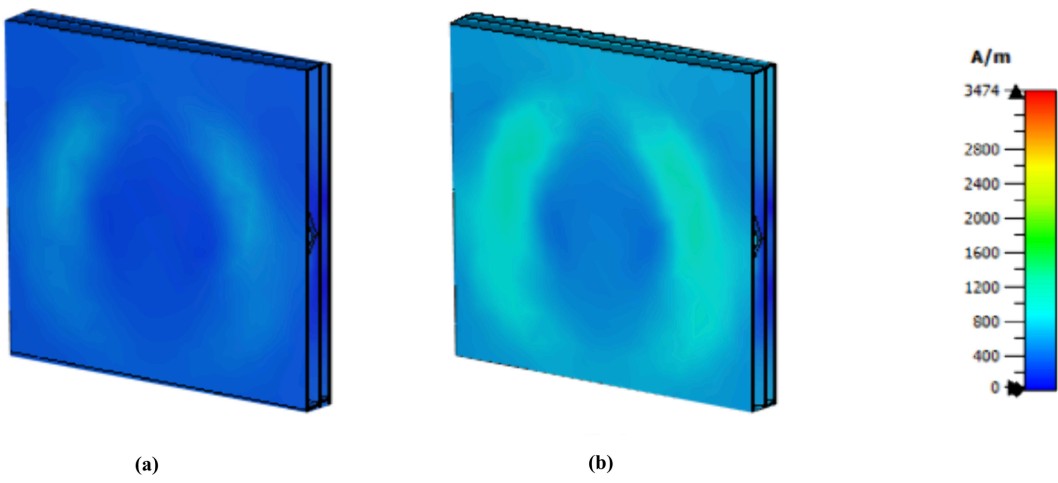

**Fig 23. H field MWI technique at 3.62 THz (a) normal cell (b) cancer cell.**

The FOM values for this sensor are 6.82, 6.95, and 5.6, which are excellent values. These values are compared with recent research in Table 5. This sensor is very sensitive for biomedical applications due to its high sensitivity and high form value. Because it performs better in the 1.34 to 1.4 refractive index region, this sensor can, therefore, be used to detect biological substances. Additionally, testing using sample-specific tests for hyperglycemia, malaria, and cervical cells amply validates its effectiveness in these areas. Given its benefits, which include its straightforward design and great sensitivity, this sensor can be applied in biological applications.

Although the current study is fully simulation-based and utilises ADS to validate the design using comparable circuit modelling, the suggested structure was created with the feasibility of practical production in mind. The absorber can be fabricated using well-known microfabrication methods, such as inkjet printing or sputtering, which are frequently

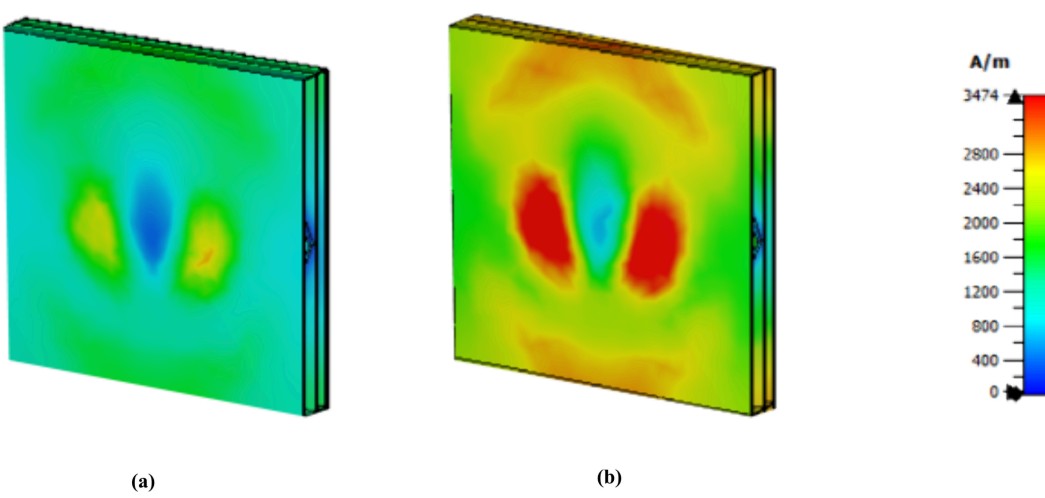

**Fig 24. H field MWI technique at 5.63 THz (a) normal cell (b) cancer cell.**

**Table 5. Comparison of figure of merit with recent research.**

| Ref. | Liu et al. [64] | Tan et al. [58] | Lu et al. [59] | Hamza et al. [51] | Proposed |
|------|-----------------|-----------------|----------------|-------------------|----------|
| FOM | 1.53, 1.57 | 2.75 | 2.94 | 0.86, 1.15 | 6.82, 6.95, 5.6 |
| Year | 2020 | 2022 | 2022 | 2023 | 2025 |

employed in the terahertz domain to create metallic resonators on both rigid and flexible substrates, as discussed earlier. After the biosensor is manufactured, its sensitivity and absorption performance can be evaluated experimentally with high-precision terahertz time-domain spectroscopy (THz-TDS) [65]. Before actual sample testing, a dry phantom setup that mimics the electromagnetic behaviour of living tissues will be utilised as a preliminary step to fine-tune sensor performance. Subsequent research will focus on utilising the sensor to detect diseases. Human sample testing will require ethical approval, and subsequent clinical trials will include statistical evaluation of diagnostic performance using sensitivity, specificity, and predictive accuracy.

## Conclusion

This work presents an octagon-enclosed unique MTM structure that operates in a 1- 6.5 THz frequency range for application in the biomedical field. The size of the layout is $41 \times 41$ $\mu m^2$ and PTFE is used as a substrate, while gold is used both front and back. This MMA produces remarkable absorption values for all three peaks, that are 99.6%, 99.6%, 99.4% for the frequencies 1.85 THz, 3.62 THz and 5.63 THz respectively. The electric field, magnetic field, and surface current were also discussed, and an equivalent circuit using ADS was developed to validate its performance. This structure also performs well in terms of sensing, with calculated values of 1.5 THz/RIU, 1 THz/RIU and 0.66 THz/RIU for these frequencies peaks. Several samples have been used to evaluate the performance of sensors in the biomedical field and MWI technology has also been used discussed for cancer cell detection. The quality factor values at frequencies of 1.85 THz, 3.62 THz, and 5.63 THz are 19.13, 25.16, and 39.13, respectively. Correspondingly, the figure of merit (fom) values for these frequencies are 6.82, 6.95, and 5.6. Because this sensor can perform well in terms of biochemical analytes, it could be

useful in biomedical applications for detecting glucose levels in blood, malaria, early-stage cancer, etc.

## Supporting information

**S1 Data. Relevant data for this study.**
(XLSX)

## Author contributions

**Conceptualization:** Asad Miah, Sams Al Zafir.

**Data curation:** Asad Miah, Sams Al Zafir, Md. Hasnain, Sayed Muhammad Anowarul Haque, Abdul Wahed.

**Formal analysis:** Joyonta Das.

**Methodology:** Asad Miah, Sams Al Zafir, Sayed Muhammad Anowarul Haque, Abdul Wahed.

**Project administration:** Asad Miah.

**Resources:** Md. Hasnain, Joyonta Das.

**Software:** Asad Miah, Sams Al Zafir, Md. Hasnain, Joyonta Das.

**Supervision:** Asad Miah, Sayed Muhammad Anowarul Haque, Abdul Wahed.

**Validation:** Asad Miah, Sams Al Zafir, Md. Hasnain, Joyonta Das.

**Visualization:** Asad Miah, Sams Al Zafir, Md. Hasnain, Joyonta Das.

**Writing – original draft:** Asad Miah, Sams Al Zafir, Md. Hasnain, Joyonta Das, Sayed Muhammad Anowarul Haque, Abdul Wahed.

**Writing – review & editing:** Asad Miah, Sams Al Zafir, Md. Hasnain, Joyonta Das, Sayed Muhammad Anowarul Haque, Abdul Wahed.

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
