## [Decision Letter · Decision Letter 0]

2 Jun 2025

PONE-D-25-25685Development of a Highly sensitive Triple-band metamaterial absorber for biomedical sensing applications.PLOS ONE

Dear Dr. Miah,

Thank you for submitting your manuscript to PLOS ONE. After careful consideration, we feel that it has merit but does not fully meet PLOS ONE’s publication criteria as it currently stands. Therefore, we invite you to submit a revised version of the manuscript that addresses the points raised during the review process.

**ACADEMIC EDITOR:**

The manuscript presents an interesting study with potential significance in the field. However, the reviewer comments must be addressed to improve the clarity, rigor, and impact of the work

We look forward to receiving your revised manuscript.

Kind regards,

Zaky A. Zaky, Ph.D.

Academic Editor

PLOS ONE

3. We note that your Data Availability Statement is currently as follows: [All relevant data are within the manuscript and its Supporting Information files]

5. Please amend the manuscript submission data (via Edit Submission) to include authors S.M. Anowarul Haque1, and Abdul Wahed.

6. Please amend either the abstract on the online submission form (via Edit Submission) or the abstract in the manuscript so that they are identical.

Reviewers' comments:

Reviewer's Responses to Questions

**Comments to the Author**

1. Is the manuscript technically sound, and do the data support the conclusions?

Reviewer #1: Yes

Reviewer #2: Partly

2. Has the statistical analysis been performed appropriately and rigorously? 

Reviewer #1: Yes

Reviewer #2: No

3. Have the authors made all data underlying the findings in their manuscript fully available?

Reviewer #1: Yes

Reviewer #2: No

4. Is the manuscript presented in an intelligible fashion and written in standard English?

Reviewer #1: Yes

Reviewer #2: No

5. Review Comments to the Author

Reviewer #1: This research article presents a metamaterial terahertz absorber for biomedical sensing applications. The paper is well organized but needs a revision before acceptance as follows;

1. The title is suitable.

2. The abstract section presents brief information about the work, it is in a suitable form.

3. The introduction section gives a comprehensive literature review, but it is advised to improve introduction section by following studies:

- https://doi.org/10.1016/j.jmrt.2021.05.031

- https://doi.org/10.1364/OME.447855

- https://doi.org/10.1117/1.OE.63.11.115102

4. Please explain how to obtain geometric design parameters.

5. Please check figures in the main manuscript, it is not seen in the manuscript file directly.

6. Please explain why this operating band was chosen.

7. The conclusion is in a good form.

8. Please mention the experimental application possibilities and future directions.

Reviewer #2: The manuscript presents a triple-band metamaterial absorber (MMA) for terahertz biomedical sensing, achieving high absorption and sensitivity. While the study is comprehensive in scope and simulation-based validation, several limitations and drawbacks should be addressed to enhance the work's scientific rigor and real-world relevance.

1- The entire study is simulation-based, using CST Studio Suite and ADS, without experimental fabrication or measurement.

2- Simulation results, while promising, do not account for real-world fabrication imperfections, material losses, or environmental noise.

3- The study focuses on refractive index sensing of a few biological targets (glucose, malaria, HeLa cells). No analysis is provided on how the sensor performs with mixed samples.

4- As we know, the given absorber is a periodic structure. How to put the test object in?

5- The device operates in 1–6.5 THz, but comparison with existing literature (Table 3, Table 4) often includes devices operating in lower or broader ranges.

6- While some angle-dependent studies are shown, the absorption shifts are noticeable beyond 30°, especially at 60°, where distortions appear.

7- How can we improve the performance of the given sensor.

8- In Fig.1, the basic of information is not clear. The physical model of materials is not given.

9- Please highlight the advance of the study in Introduction. Please explain the development and creative work. The literature review should be carefully considered.

10- Please polish the abstract. Please check the logic of abstract. Please add sentences to explain the meaning, the main points, the improvement and the promising application of the study. Plenty of detail data have given, however, in abstract, important procedures and results should be mentioned in simple manner. Please focus on the main points and the improvement of the study.

11- English in this paper needs improvement, which can make this paper more like a journal paper.

12- Many information without references.

13. The author should elaborate all results with reason.

14. For the explanation of sensors, please add the extra contents or theory to analyze.

15. It is recommended to cite the following articles:

a. https://www.nature.com/articles/s41598-023-46363-x

b. https://iopscience.iop.org/article/10.1088/2040-8986/ac8889/meta

c. https://www.sciencedirect.com/science/article/pii/S2211379720320106

d. https://link.springer.com/chapter/10.1007/978-981-97-0142-1_35

6. PLOS authors have the option to publish the peer review history of their article (what does this mean?). If published, this will include your full peer review and any attached files.

Reviewer #1: No

Reviewer #2: No

---

## [Author Response · Author response to Decision Letter 1]

13 Jun 2025

A detailed, point-by-point response to the reviewers' comments has been uploaded as a separate file titled "Response_to_Reviewers." We have addressed each comment carefully and made the necessary revisions in the manuscript.

---

## [Decision Letter · Decision Letter 1]

26 Jun 2025

Triple-band Highly Sensitive Terahertz Metamaterial Absorber for Biomedical Sensing Applications

PONE-D-25-25685R1

Dear Dr. Miah,

We’re pleased to inform you that your manuscript has been judged scientifically suitable for publication and will be formally accepted for publication once it meets all outstanding technical requirements.

Kind regards,

Zaky A. Zaky, Ph.D.

Academic Editor

PLOS ONE

Additional Editor Comments (optional):

Reviewers' comments:

Reviewer's Responses to Questions

**Comments to the Author**

1. If the authors have adequately addressed your comments raised in a previous round of review and you feel that this manuscript is now acceptable for publication, you may indicate that here to bypass the “Comments to the Author” section, enter your conflict of interest statement in the “Confidential to Editor” section, and submit your "Accept" recommendation.

Reviewer #1: All comments have been addressed

Reviewer #2: All comments have been addressed

2. Is the manuscript technically sound, and do the data support the conclusions?

Reviewer #1: Yes

Reviewer #2: Yes

3. Has the statistical analysis been performed appropriately and rigorously? 

Reviewer #1: Yes

Reviewer #2: N/A

4. Have the authors made all data underlying the findings in their manuscript fully available?

Reviewer #1: Yes

Reviewer #2: Yes

5. Is the manuscript presented in an intelligible fashion and written in standard English?

Reviewer #1: Yes

Reviewer #2: Yes

6. Review Comments to the Author

Reviewer #1: The authors have carefully addressed all the comments and suggestions provided in the initial review. The revised manuscript demonstrates significant improvement in clarity, structure, and scientific rigor. All necessary revisions have been appropriately made, and no further changes are required. I recommend the manuscript for acceptance in its current form.

Reviewer #2: I have reviewed the revised and after the author's revisions, the manuscript has been improved to an acceptable standard. I believe that it has reached the level of acceptance for publication.

7. PLOS authors have the option to publish the peer review history of their article (what does this mean?). If published, this will include your full peer review and any attached files.

Reviewer #1: No

Reviewer #2: No

---

## [Editor Report · Acceptance letter]

PONE-D-25-25685R1

PLOS ONE

Dear Dr. Miah,

I'm pleased to inform you that your manuscript has been deemed suitable for publication in PLOS ONE. Congratulations! Your manuscript is now being handed over to our production team.

Kind regards,

on behalf of

Dr. Zaky A. Zaky

Academic Editor

PLOS ONE